# Differential Functional Contribution of BK Channel Subunits to Aldosterone-Induced Channel Activation in Vascular Smooth Muscle and Eventual Cerebral Artery Dilation

**DOI:** 10.3390/ijms24108704

**Published:** 2023-05-12

**Authors:** Steven C. Mysiewicz, Sydney M. Hawks, Anna N. Bukiya, Alex M. Dopico

**Affiliations:** Department of Pharmacology, Addiction Science, and Toxicology, College of Medicine, The University of Tennessee Health Science Center, Memphis, TN 38103, USA

**Keywords:** microscale thermophoresis, patch-clamp, steroids, MaxiK channel, *KCNMB1*, vascular smooth muscle, middle cerebral artery, mouse

## Abstract

Calcium/voltage-activated potassium channels (BK) control smooth muscle (SM) tone and cerebral artery diameter. They include channel-forming α and regulatory β_1_ subunits, the latter being highly expressed in SM. Both subunits participate in steroid-induced modification of BK activity: β_1_ provides recognition for estradiol and cholanes, resulting in BK potentiation, whereas α suffices for BK inhibition by cholesterol or pregnenolone. Aldosterone can modify cerebral artery function independently of its effects outside the brain, yet BK involvement in aldosterone’s cerebrovascular action and identification of channel subunits, possibly involved in steroid action, remains uninvestigated. Using microscale thermophoresis, we demonstrated that each subunit type presents two recognition sites for aldosterone: at 0.3 and ≥10 µM for α and at 0.3–1 µM and ≥100 µM for β_1_. Next, we probed aldosterone on SM BK activity and diameter of middle cerebral artery (MCA) isolated from β_1_*^−/−^* vs. *wt* mice. Data showed that β_1_ leftward-shifted aldosterone-induced BK activation, rendering EC_50_~3 μM and EC_MAX_ ≥ 10 μM, at which BK activity increased by 20%. At similar concentrations, aldosterone mildly yet significantly dilated MCA independently of circulating and endothelial factors. Lastly, aldosterone-induced MCA dilation was lost in β_1_*^−/−^* mice. Therefore, β_1_ enables BK activation and MCA dilation by low µM aldosterone.

## 1. Introduction

Direct molecular interaction between lipids and ion channel proteins and its contribution to physiology, pathology, and drug development for therapeutics constitutes growing areas of investigation [1,2,3,4,5,6]. In particular, direct ion channel–lipid interactions have been reported to regulate the activity of Ca^2+^/voltage-gated, big conductance K^+^ (BK) channels (reviewed in [7,8,9]). In vascular smooth muscle (SM), BK channel activation and inhibition lead to SM relaxation with eventual vasodilation and SM constriction with eventual vasoconstriction, respectively (reviewed in [10]). The vascular SM BK channel phenotype is basically determined by the tetrameric association of channel-forming α subunits (encoded by KCNMA1 or Slo1 in mammals; also termed slo1 proteins) and small (two transmembrane domain; TM), regulatory subunits of β_1_ type [11,12] (Figure 1A). 

The latter is highly expressed in vascular SM while poorly expressed in most other tissues [10]. Thus, BK β_1_ suits the function of the vascular SM native channel to tissue-specific physiology; β_1_ sensitizes BK slo1 channels to subsarcolemmal Ca^2+^ levels found near the BK channel upon depolarization, which causes this channel to increase activity and thus generate outward K^+^ current that negatively feeds back on depolarization-induced SM contraction and facilitates relaxation and vasodilation [10,13,14]. 

BK β_1_ subunits also embolden the resulting heteromeric channels with distinct sensitivity to both exogenous and endogenous modulators [10,12]. In particular, β_1_ is required for BK channel activation by physiological or supraphysiological (pathological/therapeutic) concentrations of some steroids. For example, hydrophobic cholanes at concentrations reached in circulation during pathologic states that cause spillover of bile acids from the portal to the systemic circulation (10–100 M) evoke vasodilation by directly docking onto a site that is unique to the β_1_ TM2 [15,16]. In turn, µM levels of 17β-estradiol lead to BK channel activation through steroid recognition by a site different from the cholane site [17]. In addition, β subunits other than β_1_ (β_2_ and β_3_) endow heterologously expressed recombinant BK channels to increase activity in response to the androstane DHEA and the pregnane corticosterone [18]. In contrast, pregnenolone, another pregnane, inhibits BK channels and thus constricts cerebral arteries at concentrations found following therapeutic administration of pregnenolone (µM; discussed in [19]). Remarkably, pregnenolone actions on SM BK and cerebral artery diameter are independent of β subunits. Rather, α subunits suffice for steroid action [19]. Thus, whether β_1_ can endow BK channels with sensitivity to pregnane steroids remains unknown. Lastly, cholesterol at concentrations found naturally in mammalian membranes also reduces BK channel activity independently of regulatory subunits [16,20] (Figure 1A). Moreover, several Cholesterol-Recognition-Amino acid Consensus (CRAC) motifs in the slo1 channel cytosolic tail domain have been identified as mediators of cholesterol inhibitory action [21]. 

Similar to pregnenolone and corticosterone, aldosterone is a pregnane steroid (Figure 1B). Aldosterone exerts a myriad of genomic and nongenomic actions in the kidney and cardiovascular system to sustain proper homeostasis, largely through activation of mineralocorticoid receptors (MRs) [22,23,24]). Several lines of evidence, however, point to physiological or pathophysiological roles for aldosterone and MR in the brain and the cerebrovasculature: (a) presence of aldosterone synthase (CYP11B2) mRNA and protein, and functional MR in brain and cerebral blood vessels [23,25,26,27,28]; (b) patients with hyperaldosteronism suffer stroke events more frequently than essential hypertensives despite having lower blood pressure values [29,30]; (c) intracerebroventricular administration of aldosterone/MR blockers (e.g., eplerenone) alters cerebrovascular function and stroke outcome independently of changes in systemic blood pressure [24,31]. Likewise, aldosterone infusion into the fourth ventricle causes systemic effects independently of changes in systemic blood pressure and kidney function [32]. Interestingly, genetic ablation of BK channel β_1_ subunits leads to hyperaldosteronism and hypertension [33], the latter being an independent risk factor for stroke [34]. Collectively, BK channel involvement in aldosterone’s cerebrovascular actions and the participation of specific BK channel subunits remain uninvestigated. 

Combining microscale thermophoresis (MST) to determine aldosterone–BK subunit binding, patch-clamp methods on native SM BK channels, and middle cerebral artery (MCA) diameter determinations in the presence and absence of a functional endothelium in *wt* vs. β_1_^−/−^ (*KCNMB1^−/−^*) mice, this study demonstrates that aldosterone (μM) binds to both α and β_1_ subunits. However, aldosterone is only able to activate SM BK channels in the absence of β_1_ at very high concentrations (>>10 µM). However, β_1_ coexpression increases aldosterone potency to activate SM BK channels, enabling this steroid to evoke a mild but consistent MCA dilation at much lower concentrations, a steroid action that is independent of endothelial and circulating factors. 

## 2. Results

### 2.1. Aldosterone Binds onto BK α and β_1_ Subunits with Different Affinity

To address whether aldosterone may directly interact with SM BK channels and, eventually, modify their activity, we first evaluated whether aldosterone actually bound to the BK channel subunits (α and β_1_) that primarily determine the SM BK channel phenotype [10]. We chose MST because this methodology is optimal for detecting and quantifying biomolecule interactions by the thermophoretic detection of time-dependent changes in conformation, charge, and size of a molecule as they are evoked by a binding event [35]. We tested a wide concentration range of aldosterone (sub-µM–100 µM) that spanned the concentration range (1 to tens of μM) reported for other vasoactive steroids including cholanes, 17β-estradiol complexed with albumin, and pregnenolone to bind/be directly recognized by BK channels made of α ± β_1_ subunits and, thus, modify channel activity [17,19,36]. Thus, we plotted changes in fluorescence as function of temperature for aldosterone and BK α subunits cloned from cerebrovascular SM (cbv1 isoform; [37]) (Figure 2A) or BK β_1_ subunits (Figure 2C). Aldosterone binding to α subunits at very high concentrations (100 µM) is underscored by an evident leftward shift in the fluorescence–temperature plot (Figure 2A). However, to detect binding events that may occur across the whole aldosterone concentration range, we obtained two different parameters: the start of thermal-induced conformational changes (onset) or peak change (inflection) events. From these two parameters, we obtained the difference in temperature (delta, Δ) between onset and inflection to identify aldosterone binding to cbv1 (Figure 2B) or β_1_ subunits (Figure 2D). The effects of aldosterone on onset and inflection of temperature-induced conformational changes are shown in Appendix A. It is the delta parameter, however, that best reflects ligand-driven changes in thermally-driven conformational changes folding of a protein [38]. Thus, regarding BK α subunits, when compared to DMSO-containing controls (see Material and Methods), aldosterone displayed a bimodal behavior with significant changes in delta parameter at 0.3 µM and, further, concentration-dependent changes from >3 μM aldosterone, with points obtained at 10 and 100 μM being significantly different from controls (Figure 2B).

These results can be interpreted as aldosterone interacting with two binding sites in the BK α subunit of high affinity, which recognizes 0.3 M aldosterone, and of low affinity, which recognizes >3 μM aldosterone.

Regarding BK β_1_ subunits, aldosterone established binding events across all the range of aldosterone concentrations, yet statistically significant differences (*p* < 0.05) from control were only reached at 0.3 and 100 μM (Figure 2D). These data can be interpreted as BK β_1_ subunits, as their α counterparts also provide binding to aldosterone via two sites of different affinity: sub-μM to low-μM aldosterone, and ≥100 µM aldosterone. 

### 2.2. Activation of Cerebrovascular Smooth Muscle BK Channels by µM Aldosterone Requires BK β_1_ Subunits

To address the functional consequences of aldosterone differential binding to BK α and β_1_ subunits, we conducted single-channel resolution patch-clamp studies to determine aldosterone action on the steady-state activity of native channels from MCA SM cells isolated from *KCNMB1^-^*^/-^ (i.e., β_1_-missing) vs. their *wt* counterparts (C57BL/6). Each concentration of aldosterone and its corresponding DMSO-containing control solution were applied to a different SM membrane patch through bath perfusion (see Methods) (a) only after several min of patch excision from the cell, and (b) only for a few min to avoid contribution of cell metabolism to, and desensitization in, aldosterone action, respectively. The bath solution contained µM levels of free Ca^2+^ (see Methods), which are reached in the vicinity of the BK channel in cerebrovascular SM to effectively render channel activation [39]. For studies in isolated SM cells (this section) and cerebral artery segments (next section), we chose MCA because (1) this artery perfuses more cerebral territories than the other branches of Willis’ circle [40,41]; (2) MCA tone and diameter modifications are associated with numerous cerebrovascular, including ischemic, disorders [42,43,44]; (3) CA has been used in previous studies documenting aldosterone and/or MR-mediated cerebrovascular actions independently of changes in systemic blood pressure, including MCA remodeling [45,46,47]. MCA autoregulation disruption and increased BF velocity were related to increased aldosterone levels in humans [48]; (4) MCA segments and isolated SM cells were used in our previous studies documenting direct modulation of cerebrovascular SM BK channels by other steroids, including lithocholate [36], pregnenolone [19], and cholesterol [49]. Thus, BK channel recordings (time-expanded traces shown in Appendix A) obtained in *KCNMB1^−/−^* mouse MCA SM cells show that aldosterone evoked a small increase in BK NPo (10% over control) only when applied at 100 µM (Figure 3A). Moreover, averaged data show that this steroid failed to increase BK channel activity across a wide range of smaller concentrations (0.03–10 M) in MCA SM cells from *KCNMB1^−/−^* mouse (Figure 3B). Comparison of functional data in this mouse model with aldosterone binding to α subunits (Figure 2B) suggests that high-affinity binding site for aldosterone identified in these channel-forming proteins has no functional impact on BK steady-state activity. In turn, the low-affinity site for aldosterone (10–100 μM) in the BK α subunit (Figure 2B) could play some role in the aldosterone-induced minor increase in NPo (Figure 3B) observed in MCA SM from *KCNMB1^−/−^* mouse, as this aldosterone action was evoked under conditions that rule out involvement of cell integrity, metabolism, and freely diffusible cytosolic signaling molecules. At this very high, sub-mM aldosterone, however, we cannot rule out that this increase in BK NPo could be related to lipid-mediated mechanisms secondary to aldosterone insertion into the lipid bilayer [50], a topic that is beyond our current focus on direct steroid binding to BK proteins. As discussed elsewhere [51] for cholesterol, however, lipid-mediated and protein subunit binding-mediated mechanisms in steroid action on transmembrane receptors should not be considered mutually exclusive.

Figure 3C shows representative channel recordings from *wt* mouse SM cells. In these cells, which express β_1_-containing native BK channels, aldosterone reversibly increased NPo at a concentration (3 μM) that was ineffective to increase the steady-state activity of β_1_-missing BK channels under identical conditions (Figure 3B). The effect of aldosterone on *wt* BK channels was concentration-dependent, with EC_50_~3 µM and EC_100_~10 µM, at which NPo increased 20% over control values (Figure 3D). Thus, the potency for aldosterone to activate β_1_-containing BK is shifted leftward by two full log units when compared to that of their β_1_-missing counterparts (Figure 3D vs. Figure 3B). Since aldosterone was applied to I/O patches several min after patch-excision from the cell, this effect on channel steady-state activity is independent of cell integrity, metabolism, and freely diffusible cytosolic factors, including signaling downstream of MR activation. Rather, it involves aldosterone interaction with the SM BK channel and/or its immediate proteolipid environment in the native SM membrane. From comparing this electrophysiological outcome with binding data (Figure 2D), both levels of analysis show aldosterone interactions at low μM, suggesting that aldosterone binding to β_1_ leads or, at least, contributes, to aldosterone activation of β_1_-containing SM BK channels (see Discussion). 

### 2.3. Aldosterone at Concentrations Well above Physiological Levels Mildly Dilates Middle Cerebral Arteries Independently of Circulating, Metabolic, and Endothelial Factors, an Action That Requires BK β_1_ Subunits

To determine any possible impact of direct BK channel–aldosterone interactions on organ function, we addressed the effect of aldosterone on vessel diameter by probing isolated, in vitro-pressurized MCA segments in presence and absence of endothelium from both *wt* and *KCNMB1^−/−^* mice. Absence and presence of MCA endothelium was pharmacologically determined as described in Material and Methods.

Following vessel-pressurization, bath application of 10 μM aldosterone evoked a reversible dilation of de-endothelialized MCA from *wt* mice, upon which full contraction in response to depolarizing 60 mM KCl solution was routinely conducted to check for vessel contractility and, thus, viability (Figure 4C). Aldosterone action was concentration-dependent (Figure 4D), with EC_50_ and EC_100_ in the same order of magnitude/that matched those to evoke BK channel activation in *wt* mice MCA SM cells (Figure 3D). Figure 4D shows that at EC_100_, aldosterone caused a consistent but mild vessel dilation, with MCA diameter increasing 2% over presteroid values. It should be noted, however, that flow is linked to vessel radius, and thus diameter, by a fourth-power relationship [52], which will transform a ~2% increase in diameter into an ~8.2 increase in local blood flow. As found for BK channel activation, the dilation exerted by aldosterone on MCA from *wt* mice was blunted in MCA from *KCNMB1^−/−^* mice (Figure 4A,B), underscoring the functional role of regulatory β_1_ subunits in BK-channel-mediated aldosterone action on MCA SM, whether evaluated at single-channel level or on organ function. 

### 2.4. Endothelium Sustains Aldosterone-Induced Dilation of Mouse MCA

Lastly, we conducted experiments identical to those described in Section 2.3, yet we preserved the endothelial layer before applying aldosterone. In intact MCA segments from *wt* mice, aldosterone-induced dilation (Figure 5C,D) was similar to that observed in their de-endothelialized counterparts (Figure 4C,D), underscoring that the key mediators of aldosterone action (i.e., BK channels) reside in the vascular smooth muscle. Lastly, the poor (if any) efficacy of aldosterone to modify MCA diameter in de-endothelialized MCA from *KCNMB1^−/−^* mice (Figure 4A,B) was not modified by the presence of endothelium (Figure 5A,B). 

## 3. Discussion

Previous studies have documented a differential effect (activation vs. inhibition) and distinct subunit participation in the modulation of heteromeric BK channels (i.e., α ± different β types) by signaling steroids [16,18,19,36,53] These studies probed steroid action on engineered BK channels of different subunit combinations and/or used computational modeling that identified putative residues to be substituted in recombinant channels and eventually shown to affect steroid action on the channel. Thus, it has been claimed that a given steroid binds to a distinct BK subunit/site, although actual biochemical binding was not proven. A single study [17] did show that HEK cells expressing BK α + β_1_, but not those expressing α only, could “bind” (i.e., increased fluorescence) 10 μM estrogen, although the ligand under study was a fluorescein isothiocyanate-labeled 17β-estradiol covalently linked to albumin, not 17β-estradiol itself. In the present study, we used nanoscale differential fluorimetry to measure direct chemical binding between an isolated BK subunit and the steroid molecule (aldosterone) itself. This method, which detects temperature-driven unfolding of a protein based on the intensity of fluorescence that is intrinsic to Tyr, Trp, or Phe, was chosen because it does not require extra labeling of the protein, thus allowing real-time measurements of protein folding in the absence or presence of ligand of interest without interference by a label [54]. From parameter Δ measurements (see Section 2.1), our data demonstrate for the first time steroid binding (aldosterone; range under study 0.3–100 μM) to α and β_1_ subunits directly, i.e., in absence of each other, third party proteins, and membrane lipids. Our functional results (from patch-clamp electrophysiology and evaluation of MCA diameter changes in vitro), however, demonstrate that the presence of β_1_ is needed for μM aldosterone to increase the activity of native MCA SM BK channels. Most effects of aldosterone in the body are mediated through mineralocorticoid receptors (MR; members of the nuclear receptor family) [55]. However, the conditions of our experiments (isolated membrane patches excised from MCA SM cells with recordings evoked several min after patch excision) indicate that activation of β_1_-containing BK channels by aldosterone occurs independently of cell integrity, metabolism, and cytosolic signaling. This steroid action, in turn, results in MCA dilation, which is not significantly modulated by the endothelium (Figure 5D vs. Figure 4D). It has been reported that acute block of MR by spironolactone increases the potency of UTP to dilate MCA but does not alter MCA responses to SNP, which underscores the ability of MR block to reduce endothelium-dependent, but not endothelium-independent, MCA dilation [56]. Likewise, forearm venous occlusion plethysmography in humans has demonstrated that I.V. aldosterone infusion attenuates endothelium-dependent vasodilatation (by acetylcholine), whereas endothelium-independent vasodilatation is not affected [57]. Therefore, we speculate that any constriction due to aldosterone activation of endothelial MR is not able to counteract the dilation that results from steroid activation of SM BK channels. Collectively, our biochemical and functional data on isolated SM cells and MCA segments, whether intact or de-endothelialized, are consistent with the idea that aldosterone binding to β_1_ subunit proteins mediates, or at least participates in, the electrophysiological and vasoactive actions of μM aldosterone in the MCA territory of the mammalian brain vasculature. 

The differential involvement of β_1_ vs. α subunits in aldosterone modulation of BK channel function (i.e., mild activation of β_1_-missing channels at very high μM concentrations vs. activation of β_1_-containing channels by low μM) resembles the molecular pharmacology of bile acids and related cholanes on BK channels [36]. However, cholanes and aldosterone differ both in potency and efficacy to activate β_1_-containing BK channels: EC_50_ = 46 and 3 μM for lithocholate and aldosterone, respectively, while maximal increase in NPo reaches ~390 and ~125% of presteroid values for lithocholate and aldosterone, respectively ([36] vs. data shown in current Figure 3D). Noteworthy, the “cholane site” in β_1_ recognizes bean-shaped cholanes [15,58] that fit a rather convex area in TM2 defined by Thr169 and the protruding Leu171 and Leu172 [16]. The rather flat nature of the aldosterone molecule fails to stably dock onto this site, offering one plausible explanation for the quantitative differences in the pharmacological responses of β_1_-containing BK channels to aldosterone vs. lithocholate and structurally related cholanes. Consistent with this interpretation, lithocholate close analogs with a flat-shaped molecule are highly ineffective in activating β_1_-containing BK channels at concentrations < 100 μM [58]. In turn, another two steroids with flat, rather than bean-shaped, structures, cholesterol and pregnenolone (a pregnane steroid as aldosterone), do not directly activate BK channels but robustly reduce channel activity, with α subunits being sufficient for steroid action [19,20,21]. Present data show that aldosterone fails to do so (Figure 3B), highlighting the unique profile of aldosterone regarding consequences of direct BK channel–steroid interactions. Collectively, our present study underscores a heterogeneous structure–activity scenario in steroid–BK channel subunit direct interactions: (a) the final effect on BK channel steady-state activity and/or distinct subunit participation does not correlate with the number of carbons in the steroid series, and (b) a given steroid group can produce β_1_-mediated activation or α-mediated inhibition. Indeed, inhibitors working via α include cholestanes (C27; e.g., cholesterol), and activators via β_1_ include cholanes (C24; e.g., bile acids), pregnanes (C21; e.g., aldosterone), and estrogens (C18; e.g., 17β-estradiol) (Figure 1A). However, as mentioned above, pregnenolone (a pregnane steroid as aldosterone) inhibits BK channels via α subunits. 

The organ and organismal consequences of BK channel targeting by μM steroid cannot be generalized. For example, under physiological or therapeutic scenarios, it is unlikely that 17β-estradiol reaches 10 μM, whether locally or systemically [59], a concentration reported to bind and activate β_1_-containinig BK channels [17]. Under pathological conditions (e.g., severe portosystemic shunting), however, bile acids in systemic circulation do reach concentrations that activate SM BK channels and evoke vasodilation (see Discussion in [15]). In turn, the neurosteroid pregnenolone may reach sub-μM levels locally following therapeutic administration (reviewed in [19]). Regarding aldosterone in systemic serum, patients with primary hyperaldosteronism and hypertension may reach at most 8 nM aldosterone [60]. While high levels of aldosterone may be associated with decreased cerebrovascular function in hypertension, systemic aldosterone levels to evoke this effect also remain much lower than μM [48]. Likewise, in humans admitted with acute ischemic stroke, the median aldosterone serum levels is 0.4 nM [61]. On the other hand, it should be noted that interaction of BK β_1_ with μM cholanes [36] led to cholane site identification [16] in this subunit, followed by ligand-base pharmacophore design [3] and identification of nonsteroidal compounds that activated SM BK channels and thus evoked MCA dilation [19], with some activators working at nM [6]. In an analogous route, identification of BK β_1_ as a direct aldosterone functional target could lead to novel, more potent compounds of possible use to treat vascular dysfunction and eventual cognitive dysfunction associated with aldosterone actions on the brain vasculature, whether associated with hypertension or not [24,48,56].

## 4. Materials and Methods

### 4.1. Immunoprecipitation and Nanoscale Differential Scanning Fluorimetry

Chinese Hamster Ovarian (CHO) cells were purchased from Sigma-Aldrich (Sigma; 85051005; St. Louis, MO, USA). After pelleting, cells were lysed with Pierce IP lysis buffer (ThermoScientific; 87787; Rockford, IL, USA) and 1% halt-protease inhibitor cocktail, EDTA-free (100x) (ThermoScientific; 78425; Rockford, IL, USA). Low sonication was performed to remove the membrane and to prevent foaming (4 times × 5 s, performed twice at ~2 sonication power). The lysate was then centrifuged at 21,100× *g* for 15 min. Once completed, the supernatant was removed and placed on ice. Immunoprecipitation was performed using a dynabeads protein G kit (Invitrogen; 10007D; Lithuania). Equal amounts of dynabeads were added to separate tubes. Next, FLAG-tag antibody (Abclonal; AE092; Woburn, MA, USA) was bound to the aliquoted dynabeads at a concentration of 30 μg, at room temperature for 10 min. Bead antibody solution was removed, and the beads were washed with washing buffer provided by the kit. Equal amounts of lysate were added per tube so that similar amounts of protein were in each tube. Tubes were allowed to incubate on a tube revolver rotator (ThermoScientific; 88881001; Rockford, IL, USA) at room temperature for 1 h and 30 min. Then, lysates were removed, and beads were washed 3 times. After the third wash, beads were transferred to another tube to prevent potential contamination from nonspecific proteins. Washing buffer was removed, and elution buffer was added to remove protein from antibody. Then, 1M Tris-HCl (ThermoScientific; BP1757; Fair Lawn, NJ, USA) was added to the elution buffer to adjust the solution pH to 7.4. Aldosterone in DMSO (see *Chemicals* below) stock was added to each tube for a final concentration of 0.3, 1, 3, 10, or 100 μM. Identical DMSO concentrations were added to each sample to account for potential DMSO-related differences. Eluted protein was added to capillaries in triplicates for nanoscale differential scanning fluorimetry (nanoDSF). Once all eluted protein samples were added, a temperature ramp was set to 1 °C/min between 15 and 95 °C. Analysis was conducted based on onset and inflection points from the built-in Nanotemper PR Thermcontrol software. Concentration response curves (CRCs) were created and fitted with a Boltzmann function of the type $y=\frac{A1-A2}{1+e^{x-x0}}+A2$, using Origin 2022 software (OriginLab; 2023b; OriginLab Corporation; Northampton, MA, USA).

### 4.2. Isolation of Mouse Middle Cerebral Artery Segments and Individual Myocytes

Male 8–12-week-old C57BL6/J mice were purchased from Jackson Laboratories (Bar Harbor, ME, USA) and acclimated for three days upon arrival. *KCNMB1^−/−^* mice were originally gifted by Dr. Robert Brenner (University of Texas Health Science Center, San Antonio, TX, USA) and bred at UTHSC until reaching 8–12 weeks of age. Both mouse strains were housed under 12/12 day–night cycle and subjected to standard husbandry procedures according to the UTHSC Animal Care Unit. Once deeply anesthetized via isoflurane inhalation, mice were immediately euthanized via decapitation with sharp scissors. The brain was carefully removed, and MCAs were dissected out and placed onto ice-cold dissociation medium (DM) with the following composition (mM): 0.16 CaCl_2_, 0.49 EDTA, 10 HEPES, 5 KCl, 0.5 KH_2_PO_4_, 2 MgCl_2_, 110 NaCl, 0.5 NaH_2_PO_4_, 10 NaHCO_3_, 0.02 phenol red, 10 taurine, and 10 glucose. Arteries were cut into 1–2 mm long segments and placed into 3 mL of DM containing 0.03% papain, 0.05% bovine serum albumin (BSA), and 0.004% dithiothreitol at 37 °C in a clear polystyrene tube (Falcon; Serial #352054). Arterial segments were incubated and shaken in a water bath incubator at 37 °C for 10 min. Then, the tube with arteries was removed from the incubator and the supernatant was discarded. The tissue was transferred to another polystyrene tube with 3 mL of DM consisting of 0.06% soybean trypsin inhibitor, 0.05% BSA, and 2% collagenase (26.6 units/mL). This tube was vortexed and returned to the shaking incubator for additional 10 min. Lastly, the supernatant was removed after incubation, and a solution containing 3 mL of DM and 0.06% soybean trypsin inhibitor was added. The tissue was pipetted using a series of borosilicate Pasteur pipettes with decreasing internal diameter tips for obtaining individual myocytes. The solution was gently vortexed and 0.06% BSA was added. Individual myocytes (≥5 myocytes/field using a 20× objective) could be visualized using an Olympus IX-70 microscope (Olympus American Inc., Woodbury, NY, USA). The myocyte-containing DM was stored on ice, and myocytes were viable for ~4 h after isolation.

### 4.3. Electrophysiology Data Acquisition and Analysis

BK channel currents were recorded from excised I/O patches obtained from individual MCA myocytes. Bath and electrode solutions were identical and consisted of (mM) 130 KCl, 5 EGTA, 1.6 HEDTA, 2.28 MgCl_2_, 15 HEPES, and 5.22 CaCl_2_. The pH was held constant at 7.4 and was checked immediately prior to recordings. In all experiments, free [Ca^2+^] in solution was adjusted to the desired value by adding the appropriate amount of Ca^2+^ from a 1 mM CaCl_2_ stock. Nominal free Ca^2+^ was calculated with MaxChelator Sliders (Stanford University).

An agar bridge with Cl^−^ as main anion (from a stock of 1 M KCl) was used as ground electrode. Patch recording electrodes were pulled as described elsewhere [62]. After the patch was excised from the cell, aldosterone- or solvent-containing (control) solutions were applied onto the cytosolic side of I/O patches using an automated, pressurized system (Octaflow; ALA Scientific Instruments Inc.; Farmingdale, NY, USA) through a micropipette tip with an internal diameter of 100 μm. All experiments were conducted at room temperature. 

Ionic currents at single-channel resolution were recorded under voltage-clamp conditions in gap-free mode using an EPC8 amplifier (HEKA) at 1 kHz. Data were digitized at 5 kHz using a Digidata 1550B A/D converter and pCLAMP 10.6 (Molecular Devices; San Jose, CA, USA). As index of channel steady-state activity, we used the product of the number of channels in the I/O patch (N; defined as the maximal number of opening levels at Po ≤ 1) and channel open probability (Po). Drug-induced NPo changes were determined by comparing baseline NPo (i.e., NPo with the patch/cell perfused with bath solution) to the NPo from the same patch/cell exposed to the agent under study (aldosterone/DMSO) dissolved in bath solution. NPo was automatically determined using Clampfit 10.6/10.7 software (Molecular Devices).

### 4.4. Middle Cerebral Artery Diameter In Vitro Measurements

Following MCA isolation (see section above), each artery was cut into 1–2 mm long segments. Then, one arterial segment was cannulated at each end in a temperature-controlled perfusion chamber (Scintica; CH-1; Webster, TX, USA). To remove the endothelial layer of the arterial segment, an air bubble was pushed into the vessel lumen for 90 s prior to vessel cannulation. This method has been shown to adequately remove the endothelial layer based on the differential response of treated arteries to endothelium-dependent vs. endothelium-independent vasodilators [63,64].

Utilizing a Dynamax RP-1 peristaltic pump (Rainin Instr., Columbos, OH, USA), the chamber housing the cannulated arterial segment was perfused at a rate of 3.75 mL/min with physiologic saline (PSS, mM): 119 NaCl, 4.7 KCl, 1.2 KH2PO4, 1.6 CaCl_2_, 1.2 MgSO_4_, 0.023 EDTA, 11 glucose, 24 NaHCO_3_. PSS was maintained at pH 7.4 with a mixture of 21/5/74% O_2_/CO_2_/N_2_ and maintained at 35–37 °C. Arteries were monitored with a charge-coupled camera (ImagingSource, Serial#: 06010529, Breman, Germany and ImagingSource, 36110078, Breman, Germany). The external wall diameter was measured as by us [19,65,66] and others [67,68]. This was conducted automatically using the edge-detection function of IonWizard software (IonOptics, Westwood, MA, USA), and data were digitized at 1 Hz. A pressure transducer (Living Systems Instrumentation Model PM-4; Serial #: 20-1029A; Serial #: 12-0907A; Webster, TX, USA) was used to monitor the intravascular pressure by elevating a reservoir filled with PSS. Arteries were first pressurized to 10 mmHg for 10 min to equilibrate to changes in pressure. Then, intravascular pressure was increased to 60 mmHg to induce development of myogenic tone. This was visually confirmed by artery constriction (Appendix A). Aldosterone was added after the arterial segment fully developed tone (see *Chemicals* below), and measurement of aldosterone effect was conducted only after the MCA segment reached a maximal, steady-state response to the steroid, upon which washout with PSS was immediately perfused. After washing, maximal contractility was determined with a depolarizing 60 mM KCl solution (mM): 63.7 NaCl, 60 KCl, 1.2 KH_2_PO_4_, 1.2 MgSO_4_, 0.023 EDTA, 11 glucose, 24 NaHCO_3_, and 1.6 CaCl_2_ (Appendix A). KCl was equilibrated at pH 7.4 with a 21/5/74% mix of O_2_/CO_2_/N_2_ and maintained at 35–37 °C.

### 4.5. Chemicals

Aldosterone (A9477; St. Louis, MO, USA), calcium chloride (C3881; St. Louis, MO, USA), dimethyl sulfoxide (DMSO; D8418; St. Louis, MO, USA), sodium bicarbonate (S6014; St. Louis, MO, USA), Phenol Red (P4633; St. Louis, MO, USA), and Taurine (T8961; SLBP8056V; St. Louis, MO, USA) were obtained from Sigma-Aldrich. BSA (Jackson ImmunoResearch; 001-000-162; 153078; West Grove, PA, USA), papain (MP Biomedicals; 100921; S6808; Solon, OH, USA), and soybean trypsin inhibitor (Sigma; T9129; SLCJ5327; St. Louis, MO, USA) were stored at 4 °C. Collagenase (Sigma; C8051; 118M4030V; St. Louis, MO, USA) was stored at −20 °C. Sodium chloride (S271), potassium chloride (P217), potassium phosphate (P285), and magnesium chloride (M33) were purchased from Fisher Scientific (Fair Lawn, NJ, USA). Magnesium sulfate (MX0070-1) was purchased from EMD Millipore Sigma (Gibbstown, NJ, USA). Ethylenediamine tetra-acetic acid (EDTA) (0.5M; E177) was purchased from VWR (Solon, OH, USA). Clinical blood gas mixture (5.0% carbon dioxide, 21% oxygen, and 74% nitrogen; UN1956) was purchased from Nexair (Memphis, TN, USA). A stock solution of aldosterone was first made (either 1 mM or 10 mM aldosterone) in DMSO. Aliquots were made and frozen at −20 °C until needed, and aliquots were only kept up to a year. The DMSO–aldosterone mixture was added to 30 mL of physiologic saline for a final concentration of (µM) 0.03, 0.3, 1, 3, 10, and 100. Aldosterone-containing DMSO was added to physiological saline immediately prior to profusion. The saline mixture was covered with parafilm to prevent any evaporation.

### 4.6. Statistical Analysis

Statistical analysis was performed using SPSS v27. When the number of observations in the groups under comparison exceeded 6, and the Gaussian distribution of the data was confirmed by the Kolmogorov–Smirnov test, analysis was performed using an unpaired Student’s t-test. In all other cases, statistical analysis was conducted using the Mann–Whitney nonparametric test. For comparison of multiple experimental groups, the Kruskal–Wallis test with Dunn’s post-test were used. In all cases, testing assumed two-tailed *p* values with significance set at *p* < 0.05. Validity of the Boltzmann fit results were given by the corresponding reduced chi-square and R-square values; the reduced chi-square statistic was used to represent the goodness-of-fit testing and the R-square results evaluated the scatter of data points around the fitted regression line.

## 5. Conclusions

Our study documents, for the first time, direct binding of a signaling, circulating steroid molecule (aldosterone) to the subunits that primarily define the phenotype of SM BK channels, and advances the idea that regulatory β_1_ subunits allow μM aldosterone to evoke activation of vascular SM BK channels and endothelium-independent dilation of MCA. These findings constitute a first step towards identification of an aldosterone-recognition site in BK β_1_-subnits and eventually obtaining novel, more potent compounds that, through BK channel activation, could modify aldosterone-mediated cerebrovascular pathophysiology.

## Figures and Tables

**Figure 1 ijms-24-08704-f001:**
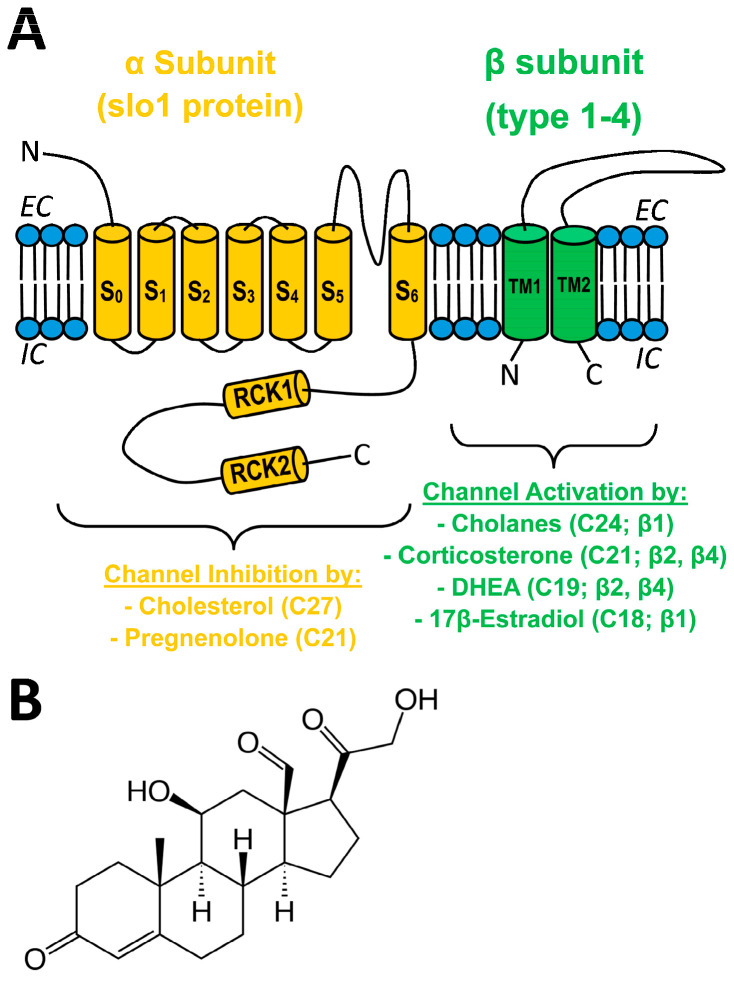
(**A**) Sketch of a BK channel heterodimer of α (slo1) (orange) and generic β (types 1–4) (green) subunits, with pore in the former between S5 and S6. Four heterodimers make the functional BK channels expressed in most mammalian tissues. The effect of different steroids on BK channel activity is shown, with the number of carbons in the steroid molecule and the BK subunit involved in steroid action in parenthesis. EC: extracellular medium; IC: intracellular medium S: transmembrane segment in α; RCK: regulator of conductance for K^+^ domain; TM: transmembrane domain in β; DHEA: dehydroepiandrosterone. (**B**) Structure of aldosterone (C21).

**Figure 2 ijms-24-08704-f002:**
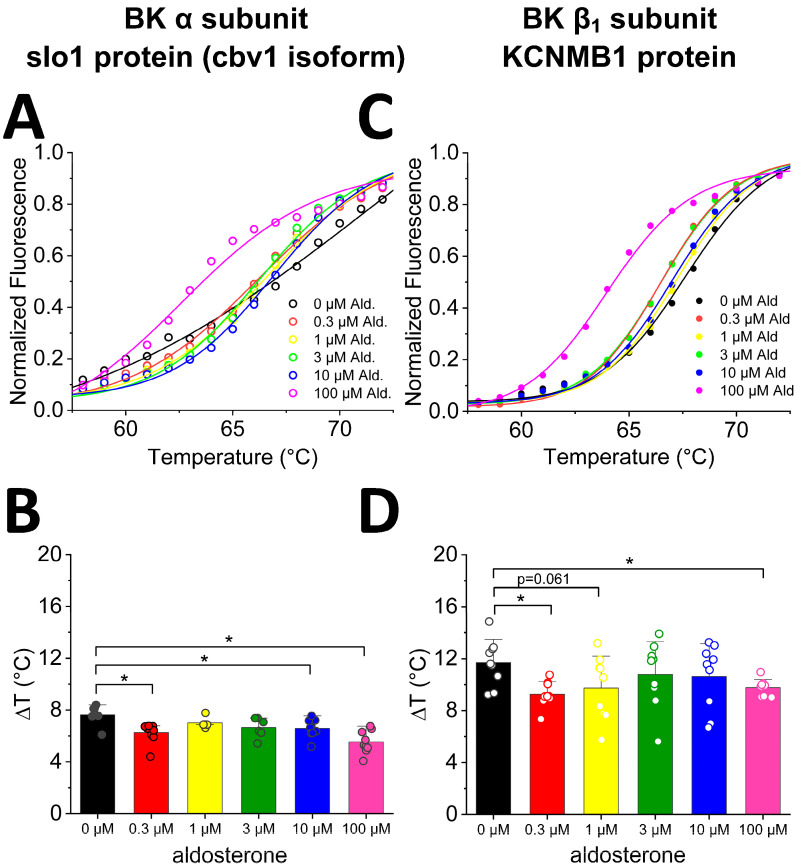
Differential scanning fluorometry on isolated BK-associated proteins shows differential binding of aldosterone. (**A**) Example of normalized traces from a single thermal unfolding run of isolated BK α subunit (slo1; cbv1 isoform). In each run, protein samples were loaded in parallel into multiple capillaries with various aldosterone concentrations. (**B**) Differences between onset and inflection point of averaged cbv1 curves show significant differences in unfolding at 0.3, 10, and 100 µM aldosterone. Average values reflect data from no fewer than three runs representing biological replicates. (**C**) Example of normalized traces from a single thermal unfolding run of isolated BK β_1_ subunit. As in (**B**), during one run, protein samples were loaded in parallel into multiple capillaries with various aldosterone concentrations. (**D**) Differences between onset and inflection point of averaged β_1_ curves show a significant difference at 0.3 and at 100 µM aldosterone. * *p* < 0.05 unless otherwise stated. Average values reflect data from no fewer than three runs representing biological replicates.

**Figure 3 ijms-24-08704-f003:**
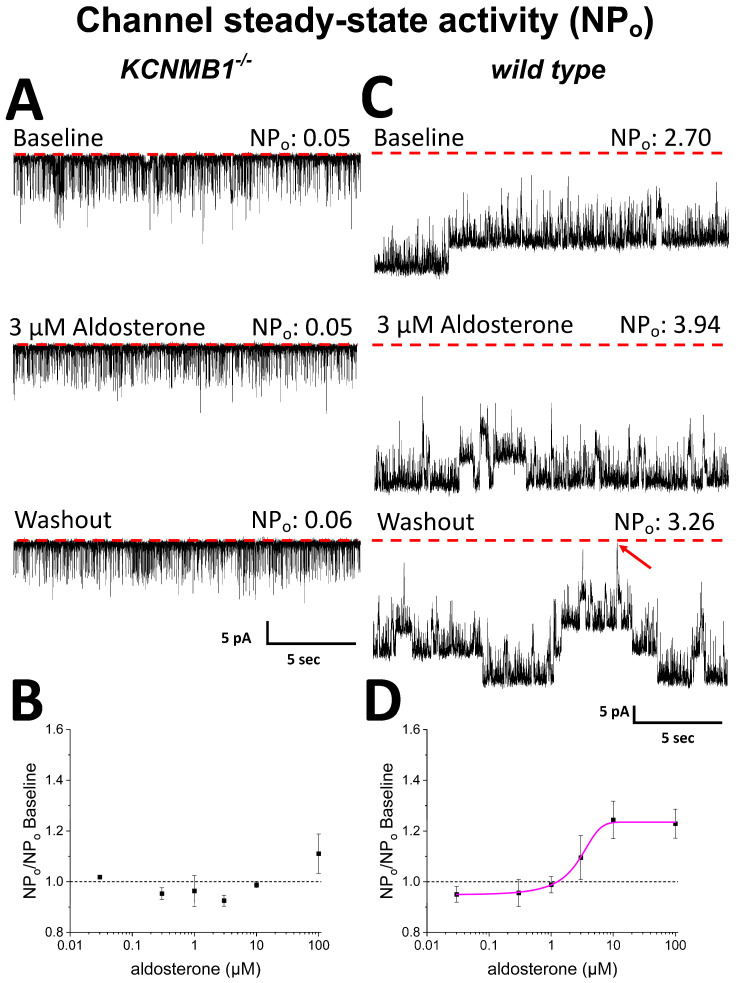
Aldosterone activates smooth muscle BK channels from *wt* mouse middle cerebral artery (MCA) in a concentration-dependent manner; this action is blunted in myocyte BK channels from *KCNMB1^−/−^* mice. (**A**) Representative channel activity traces before (top), during (middle), and after (bottom) application of 3 µM aldosterone to the cytosolic side of inside-out (I/O) patches from *KCNMB1^−/−^* myocytes. (**B**) In these myocytes, channel steady-state activity (NPo) as a function of aldosterone concentration shows a mild increase only at 100 µM. (**C**) Channel activity traces before (top), during (middle), and after (bottom) application of 3 µM aldosterone to the cytosolic side of I/O patches from *wt* myocytes. (**D**) In *wt* cells, aldosterone action is concentration-dependent: EC_50_ = 3 µM; EC_MAX_ = 10 µM. In (**A**,**C**): red dotted lines and an oblique red arrow indicate the baseline (all channels closed).

**Figure 4 ijms-24-08704-f004:**
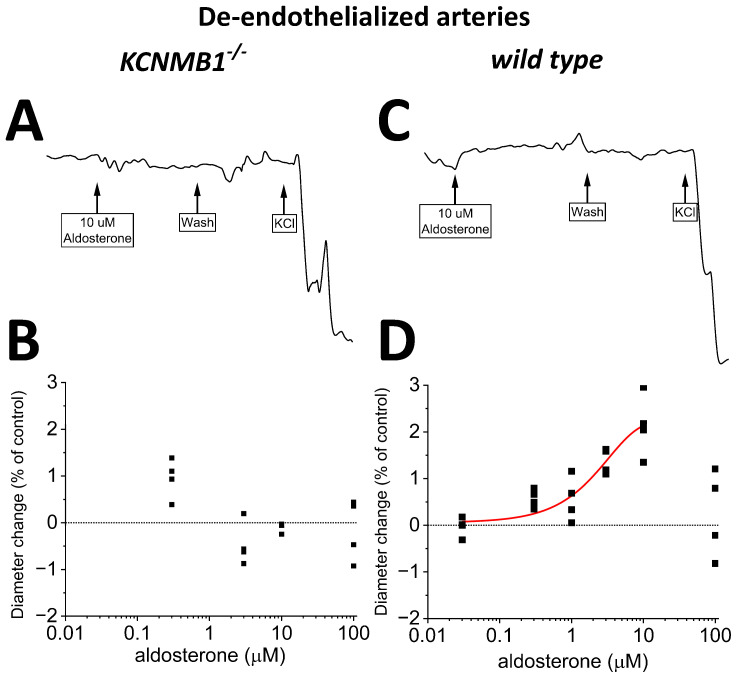
Aldosterone mildly dilates in vitro pressurized, de-endothelialized middle cerebral arteries (MCA) from *wt* mice but fails to do so in MCA from *KCNMB1^−/−^* mice. (**A**,**B**) Representative MCA diameter traces from *KCNMB1^−/−^* mouse showing diameter changes in response to aldosterone or depolarizing 60 mM KCl. (**C**,**D**) Corresponding results from *wt* mice show that aldosterone mildly dilates MCA in a concentration-dependent manner.

**Figure 5 ijms-24-08704-f005:**
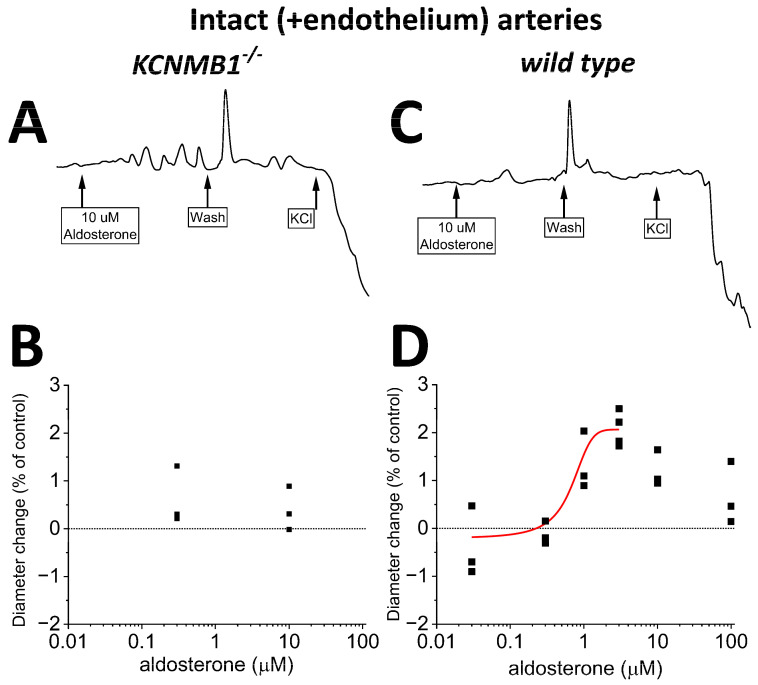
In intact (+endothelium) middle cerebral arteries (MCA) from *wt* mice, aldosterone-induced dilation is reduced when compared to that in de-endothelialized MCA (Figure 4). (**A**,**B**) Representative MCA diameter traces from *KCNMB1^−/−^* mouse showing diameter changes in response to aldosterone or depolarizing 60 mM KCl. (**C**,**D**) Corresponding results from *wt* mice show a very small dilation by >0.3 μM aldosterone.

## Data Availability

The Appendix A presented in this study are available here: https://doi.org/10.6084/m9.figshare.22795991.v1.

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
