# Peer review of "Differential Functional Contribution of BK Channel Subunits to Aldosterone-Induced Channel Activation in Vascular Smooth Muscle and Eventual Cerebral Artery Dilation"

_ijms, 2023, doi:10.3390/ijms24108704_

Round 1
Reviewer 1 Report
RE: Manuscript ID: CS-2022-0700
The study by Mysiewicz and colleagues evaluates the contribution of a and b1 subunits of BK channels to aldosterone-induced channel activation in vascular smooth muscle and cerebral artery dilation. The authors discussed the new findings and concluded that b1 subunits enable aldosterone activation of BK channels and middle cerebral artery dilation.
The authors used a rigorous approach and analysis to quantify the contribution of a and b1 subunits to aldosterone-induced BK channel activation in vascular smooth muscle and MCA dilation. The findings included a b subunit-dependent leftward shift of aldosterone-induced BK channel activation and MCA dilation. The results are novel, and the discussion is appropriate.
Major concern
It is unclear that the b1 subunit binds aldosterone at a single site (0.3-1 mM, Fig. 2). Fig.2 suggests a second low-affinity site at 100 mM. Therefore, there is no differential BK channel subunit contribution on the effects of aldosterone.
Minor concerns
1. Fig.2 Please include a1 CBV1.
2. Section 2.2. Please rewrite this section. In addition, 0.3 M was not tested in the study.
3. Line 139. Pane D, no statistical difference at 1 mM aldosterone. Please clarify.
4. Figure 3, panel C. Please provide a parallel presentation of current traces as in panel A.
5. Figure 4, panel D. Please provide data points as in panel B.
6. Provide a rationale for the use of male mice.
Author Response
Please see atttached pdf.

Reviewer 2 Report
The goal of this study was to investigate the effect of aldosterone on Calcium/voltage-activated potassium channels (BK) expressed with or without the beta-1 regulatory subunit which is specific of smooth muscle cells. The authors evaluated the aldosterone binding to the BK channel or the beta-1 subunit using microscale thermophoresis. Then, they showed that this binding effect produce the BK channel activation in middle cerebral artery (MCA) smooth muscle cells obtained from wild-type mice. However, the aldosterone failed to activate the channel in the same cells obtained from beta-1 KO mice. Finally, they showed that aldosterone produces a mild dilation of deendothelized MCA. The hypothesis is interesting, and the methodology is adequate, however, the results are poorly presented and there are some statistical concerns that should be addressed:
Major points:
1- Figure 2: It is not clear what the delta parameter represents and how was obtained in MST experiments. What is the advantage of this parameter with respect to the left shift of the curve?
Why BK channel alfa-subunit was not included in the beta-1 binding assay? Is there any technical issue? The binding of aldosterone to the alfa-subunit could be modified in the presence of beta-1 and vice versa.
The results show a similar pattern for cbv1 and beta-1: a reduction in delta parameter at low (0.3 µM) and high concentration (100 µM). However, the authors conclude that aldosterone has a different binding profile between the two subunits, showing two binding sites for cbv1 and only one for beta-1
Line 372: ” Concentration-response curve (CRC) were conducted and fitted with a Boltzmann function” but in Fig. 1 it seems that the comparison was done by one-way ANOVA.
2- Figure 3: The baseline activity of the BK channel in MCASMC from wild-type mice is too high. The activation could be underestimated since the channel is mostly activated. In this case, it may be adequate to reduce the intracellular Ca2+ concentration.
It will be good to provide more expanded recordings to evaluate if the fast opening transitions of the BK channels expressed without beta-1 are completely solved, even more, when a 1 kHz filter was applied and the sampling rate was low (5 kHz) for a fast gating channel.
3- Figure 3B, 4B, 5B, and 5D: There is no sense to fit with Hill equation data that does not show a sigmoidal relationship. Moreover, two or three concentrations tested are insufficient to evaluate a concentration-response relationship.
Reviewer 3 Report
This manuscript describes studies evaluating the possible activation of BKCa channels by aldosterone using nanoscale differential scanning fluorimetry to detect interactions of aldosterone with isolated BKCa channels +/- beta1 subunits, BKCa single-channel recording from inside-out membrane patches from wild type and beta1-knockout mice and pressure myography of MCAs from wildtype and beta1-knockout mice to assess the ability of aldosterone to induce vasodilation. The only significant concerns are as follows. First, the authors never state how much pressure-induced myogenic tone was present in their various preps used for pressure myography. This information is required to properly interpret the very small dilation induced by aldosterone. Second, he concentrations of aldosterone required to see effects are supraphysiological (an understatement) and the authors present no evidence that the binding they studies is specific for aldosterone.
Round 2
Reviewer 2 Report
The revised version of the work presented by Mysiewicz et al. presents significant improvements regarding the clarity of the data presented. Some of this reviewer's concerns have been answered and clarified, however others need further improvement.
Major points:
1- The binding curve of aldosterone on the BK expressed with the beta-1 subunit should be included. All the functional characterization of this interaction was performed by the patch-clamp technique and the measurement of arterial diameter on cells that express the whole complex, not only the beta-1 subunit.
2- According to the authors' response, the delta T parameter would be an indirect indicator of changes in the slope of the fluorescence vs. temperature curve. However, the effect of aldosterone is observed as a left shift of the curve. This is clearly simpler to quantify at the inflection and onset points than by the change in the slope itself. It is also strange that delta T has values greater than 10°C when the entire curve is defined in that same temperature range. This reviewer understands that the delta T parameter was satisfactory with other ligands, but for this ligand-protein pair, the onset or inflection point may be more appropriate.